# Environmental Effects of the COVID-19 Pandemic: The Experience of Bogotá, 2020

**DOI:** 10.3390/ijerph19106350

**Published:** 2022-05-23

**Authors:** Jeadran Malagón-Rojas, Daniela Mendez-Molano, Julia Almentero, Yesith G. Toloza-Pérez, Eliana L. Parra-Barrera, Claudia P. Gómez-Rendón

**Affiliations:** 1Instituto Nacional de Salud, Bogotá 111321, Colombia; damemo27@gmail.com (D.M.-M.); jalmentero@ins.gov.co (J.A.); ytoloza@ins.gov.co (Y.G.T.-P.); elipabarrera@yahoo.es (E.L.P.-B.); 2Doctorado en Salud Pública, El Bosque University, Bogotá 110121, Colombia; gomezclaudiap@unbosque.edu.co

**Keywords:** COVID-19, SARS-CoV-2, global environment, solid waste, air quality, transport

## Abstract

During the novel coronavirus disease (COVID-19) pandemic, several environmental factors have influenced activities and protection policy measures in cities. This has had a major effect on climate change and global environmental catastrophe. In many countries, the strategy of closing various activities such as tourism and industrial production stopped normal life, transportation, etc. This closure has a positive impact on the environment. However, the massive use of masks and personal protection could significantly increase pollution worldwide. The impact on the environment needs to be calculated to have information for public health actions. In this study, we present a first overview of the potential impacts of COVID-19 on some environmental matrices in Bogotá, Colombia.

## 1. Introduction

The worldwide spread of severe acute respiratory syndrome (SARS-CoV-2) has had a relevant impact on economies, societies, road traffic, tourism, and human interaction [1]. Since COVID-19 was declared a pandemic disease by the WHO on 11 March 2020, strategies have been focused on individual lifestyles protection, suggesting individual isolation, confinement, and introducing behavioral practices such as regular hand washing, and the use of a face mask to help prevent viral transmission, particularly in public spaces [2].

The adoption of measures for preventing COVID-19 transmission has impacted the environment in many ways. Some authors have postulated that the COVID-19 pandemic is an opportunity to “reset” the current practices that are putting in check the planetary sustainability [3,4,5]. In this vein, a better understanding of pro-environmental behaviors and practices that took part during the global lockdown may contribute to improving health globally [6].

In addition, it seems that the imposed restrictions on mobility have drastically modified patterns and personal behaviors toward energy consumption [7,8,9]. Additionally, the COVID-19 pandemic decimated tourism and business travel, as well as labor migration [10], and modified the patterns of city mobility, increasing the usage of individual transport modes and behaviors toward bike and car usage [11,12,13]. In addition, pandemic restrictions have positively impacted indicators of air and water quality and noise pollution [14], and environmental noise has been reduced by 75%. This may be associated with a notable decrease in vehicle traffic, social mobility, international trips, and the temporary closure of factories [15]. In addition, some authors have stated that countries in which a social confinement strategy was applied to stop the spread of coronavirus infection showed a notable decrease in pollution and the emission of greenhouse gases [16,17,18]. This coincides with reports of a significant reduction in carbon emissions in highly industrialized countries [19]. Nevertheless, other authors have claimed that the potential benefits of the quarantine period have been eclipsed by lifestyles based on the increase in the use of single-use products, such as personal protective equipment, thwarting efforts toward reducing plastic pollution [20,21,22].

Some reports have indicated that the production of solid waste has increased four times [14,23]. Nevertheless, there is still not enough evidence of the effects on the generation of solid waste and the generation of environmental and air pollutants during the current pandemic. It is necessary to consider the negative impacts of the pandemic in terms of global warming and hospital and general waste to formulate policies regarding waste management and environmental pollution [24]. Understanding how the lockdown and social distancing measures have impacted the environment may inspire ideas supporting the goals of achieving more sustainable cities and communities in 2030 [25]. Therefore, this study is intended to expose and evaluate the environmental impact of the COVID-19 pandemic in 2020 in Bogotá, Colombia.

## 2. Materials and Methods

### 2.1. Study Area

Bogotá, the capital of Colombia, was selected as the study area. By May 2021, the city had more than 800 thousand infections since the beginning of the pandemic [26]. Bogotá has approximately 7.2 million inhabitants. The city is located at an average of 2625 m above sea level and is in the center of Colombia. It has a length of 33 km from south to north and 16 km from east to west (Figure 1). The economy is mainly based on the service sector, commerce, manufacturing, and construction. Additionally, it is one of the most polluted cities in Latin America. Pollution is mainly derived from diesel fuel, natural gas, industrial pollution, and the destruction of forests [27]. By 2019, 2.4 million vehicles were circulating in Bogotá, among which 50% were automobiles, 20% were motorcycles, 14% were vans, and 5% were public transport vehicles [28].

During the first year of the pandemic, Colombia reported almost 1.6 million confirmed cases and roughly 42,000 deaths related to COVID-19 [29]. The national lockdown was established between 25 March and 15 April, followed by a slow increase in mobility until August 2020 [29]. Since September 2020, COVID-19 mitigation strategies were fundamentally based on promoting remote work, the usage of personal protective equipment, and preventing social contacts, such as closing schools and universities. Additionally, the Ministry of health’s guidance included social distancing, case isolation, and shielding to limit community-level transmission of SARS-CoV-2 and protect vulnerable groups.

In the specific case of Bogotá, community mobility was restricted based on the rates of occupancy of intensive care units (ICUs), through the strategy “*pico y cédula*”, which allowed mobility in the city and access to banks, supermarkets, and public transport based on the ending number of ID cards [30].

Similar to what has been observed in other scenarios, the SARS-CoV-2 epidemic in Colombia and Bogotá has been highly heterogeneous spatially. While some municipalities experienced explosive early spikes, followed by periods of very low transmission despite the near absence of NPI, many others experienced several moderate spikes interspersed with plateaus of sustained transmission [29,31].

### 2.2. Study Design

A retrospective ecological study was designed. The observed period was January 2019 to June 2021. To select the sources of information, we considered a matrix of environmental factors that may be affected by human activities [32]. In Appendix A, the environmental factors included in the study are listed, which were considered based on the availability and quality of the data and the relevance and strategic importance of the provided information.

### 2.3. Solid Waste Data Analysis

To analyze changes in the generation of solid waste in Bogotá, two sets of data were analyzed. The first set corresponds to the solid waste disposed of in the capital’s sanitary landfill. For this analysis, a time series was made comparing its generation in 2019 with that in 2020. For the second dataset, the hospital waste in the biosanitary category obtained from the Special Administrative Unit of Public Services (UAESPs) was analyzed. In this study, the generation of solid waste in kg/month by large producers (LPs), medium producers (MPs), small producers (PPs), and micro-producers type A (MA), type B (MB), and type C (MC) was determined. We included information from all companies (n = 12) who are in charge of solid waste collection in the city (Appendix A).

### 2.4. Air Quality

Secondary data from the Bogotá Air Quality Monitoring Network (RMCAB) were used to analyze changes in the concentration of criteria pollutants during the pandemic period. Additionally, a time-series analysis was performed for criteria pollutants carbon monoxide (CO), nitrogen dioxide (NO_2_), and particulate matter less than 10 microns in diameter (PM_10_) and less than 2.5 microns in diameter (PM_2.5_). Information from 11 stations operating during the 2 years was used.

Additionally, we included data from the Air Contamination And Health Effects in Microenvironments in Bogotá (ITHACA) study related to questions regarding the perception of air quality of people on their way to work or study and the personal protective elements they use to travel [33,34]. In this study, we used the data from 1821 citizens (more details are provided in Appendix A). The data were collected between February 2019 and June 2021.

### 2.5. Water Resources

Information on physical and chemical profiles of water for human consumption reported by Bogotá Aqueduct Company was included. Values are reported for turbidity in UNT, the concentration of manganese, organic matter, ammonium, dissolved oxygen in mg/L, and conductivity in µS/cm. The values are averages and maximums recorded for 2019 and 2020.

### 2.6. Transport

To analyze changes in transportation dynamics, the survey conducted in the ITHACA project was used. The survey records participants’ answers to questions associated with their transportation and air quality during the COVID-19 pandemic (n = 1821).

In addition, information on the number of trips in the city’s BRT system (Transmilenio S.A.) was collected. These values report the number of entries in three modes of transport that are part of the system. The first refers to BRT buses that run on the main lines of the city, the second mode of transport is zonal buses that run in mixed lanes, and the third mode is dual buses.

### 2.7. Statistical Analysis

For quantitative variables, averages and standard deviations were estimated. For categorical variables, frequencies were obtained. Location measures, such as quartiles, were used for some variables.

A time-series visual analysis was performed for air quality and transport (urban trips and air trips) by month and year. To establish differences between months and years, a bivariate analysis was performed using a *t*-test or Mann–Whitney U test for continuous variables and Chi^2^ test for discrete variables, where *p* > 0.05 indicated a significant difference. In addition, the data from 2020 were divided into four phases: (1) baseline (1 February 2019 to 24 March 2020); (2) strict national lockdown (25 March to 26 April); (3) first relaxation (27 April to 31 May); and (4) gradual economic opening (from 1 June onwards) (Appendix A).

The analysis was performed using R version 4.1 (Vienna, Austria) and Wolfram Mathematica version 12.0 (Champaign, IL, USA).

## 3. Results

### 3.1. Solid Waste

A decrease (33%) in disposed waste was observed in April 2020, compared with that in 2019. Nevertheless, after performing a temporal series analysis, it was observed that disposal quantities returned to their recurrent trend after the first measures of the pandemic strategy were removed (Appendix A).

It was found that large producers of biosanitary waste and micro-producers A and B significantly increased their waste generation (Figure 2), with large producers presenting an increase from 67,000 kg in March to 89,000 kg in August 2020 and micro-producers A and B presenting a 100% increase in solid waste from March to July 2020.

For LPs, an increase in biosanitary waste generation was observed between April and August 2020. For MPs and PPs, a small increase was also observed, but this coincided with the economic reopening of the sites where these types of waste are produced, which presented the largest figures for 2019. Concerning MA, MB, and MC generators, increases were observed between April and August 2020, corresponding to normalization in the generation of biosanitary waste.

The Mann–Whitney U test showed significant differences between total waste production in 2019 and 2020 (*p*-value < 0.05). Nevertheless, the median amounts of biosanitary waste in 2019 and 2020 were not different (*p*-value > 0.05) (Appendix A). When comparing only the LPs, large differences were found (*p* value < 0.05).

### 3.2. Air Quality and Citizens’ Perception

Regarding the data provided by the Bogotá Air Quality Monitoring Network, there was a decrease in criteria pollutants, such as carbon monoxide (CO), nitrogen oxides (NO_2_), particulate matter less than 10 microns in diameter (PM_10_), and fine particulate matter (PM_2.5_). For the strict quarantine stage, there was an average reduction of ~51% for CO and ~61% for NO_2_ (Figure 3a,b). These reductions were due to the decrease in transportation and halt of activities that produce fixed emissions.

Regarding particulate matter concentrations, there was a reduction of ~36% for PM_10_ and ~19% for PM_2.5_ (Figure 3c,d). It is important to highlight that the strict lockdown began at a time when the city presented critical air pollution conditions due to the atmospheric conditions of the capital. Between February and March, there are usually air pollution alerts that exceed the maximum values allowed by the WHO, as shown in the timeline for 2019.

The *t*-test and Mann–Whitney U test showed significant differences in CO, NO_2_, and PM_10_ concentrations between 2019 and 2020 (*p*-value < 0.05) (Appendix A).

In addition, it was found that the perception of air quality improved significantly during the pandemic period compared with 2019 (Chi^2^ = 25.73; *p* = 0.00001) (Appendix A). Air quality perceptions are related to the intention to use personal protective equipment (PPE) (*p*-value = <0.001, X-squared = 20.0). This relationship is evident only in surveys conducted during the pandemic. We found that before the pandemic, the use of PPE was not significant. The pandemic modified this trend and promoted the use of PPE, leading to the emergence of similar trends in the perception of the relationship between air quality and health effects (Appendix A).

### 3.3. Transport

From the time-series analysis of the data provided by Transmilenio S.A., declines of 70–80% in revenues to the city’s BRT system were observed between April and June, compared with the data of 2019 (Appendix A). This decline had not returned to habitual levels by December 2020, showing 37% fewer revenues than the previous year. The figures show the effect of the strict quarantine and are consistent with the reduction in air pollutant emissions.

From the responses obtained from the participants of the ITHACA survey, changes in the choice of a preferred mode of transportation were found (Figure 4). There was a tendency to decrease the use of public transport, such as BRT and conventional buses, along with eight-point and one-point increases in the choice of private transportation by car and by motorcycle, respectively. There was also an increase in the use of active transport modes, reaching almost four points for walking and approximately five points for biking.

### 3.4. Water Resources

When analyzing water quality according to the reports generated by the Bogotá Water and Sewerage Company, significant reductions were observed in the different pollutants analyzed. In the case of organic matter, a reduction of approximately 7 mg/L was observed in 2020 compared with 2019. Additionally, ammonium was reduced by 38%, manganese was reduced by 75.8%, turbidity was reduced by 46%, and conductivity was reduced by 49%. These values refer to the maximum concentrations observed for the two years of analysis and represent a significant decrease and, thus, favorable conditions in the city for 2020. This effect may be associated with the decrease in discharges from industries located upstream of the plant’s catchment point, i.e., in the upper basin (Appendix A).

## 4. Discussion

To our knowledge, this is the first study carried out about a city in Latin America exploring the effects of the COVID-19 on environmental matrices, including effects on water, air quality, mobility patterns, and waste production. Some studies explored descriptively the effects of the pandemic on air quality in Argentina, Chile, Colombia, and Mexico [35,36,37,38], while others reported the effects of deforestation patterns in Brazil and Peru [39]. Nevertheless, information related to solid waste production and water quality is scarce.

The restrictions established by the government generated a change in the dynamics of the city. The strict quarantine led to the closure of many places with economic activities that produce waste, atmospheric emissions, and water pollution and to an increase in solid biosanitary waste. The decrease in different environmental effects was reflected in several scenarios, generally having positive effects on the environment.

First, there was a 32% reduction in solid waste in comparison with the normal amount produced in the most critical month. However, this reduction was evident only until the gradual economic opening, at which time the tons of solid waste arriving at the landfill corresponded to the average values of previous years. In addition, there was a significant increase in the generation of biosanitary waste from large producers, from 600,000 tons per month to almost 900,000 tons/year. Some authors consider that waste production has risen as a result of COVID-19, although these changes do not follow the same pattern in different areas [40]. The increase in waste production is explained in two ways: first, disease prevention or treatment activities (hospital and lab PPE) [41] and, second, the effects of the pandemic on lifestyles, such as increased in-home cooking and online shopping. Additionally, as a result of the massive usage of PPE, this waste category substantially increased, between 18% and 425% [42,43].

On the other hand, atmospheric pollution greatly benefited from the city’s mobility restrictions. The Colombian capital tends to present atmospheric conditions that are not characterized exclusively by local emissions, as the influence of regional pollution has been demonstrated [44]. Despite this influence, mobility and industrial activities that are usually critical for the city slowed down for months. Authors have reported similar results in different regions. A study carried out in 10 countries in North America, Europe, and Asia (n = 9394) observed that a reduction in pollutant concentration was perceived, although to different extents, by all populations [45]. According to the authors, except for participants from China and Norway, participants from all countries perceived a drop in the air pollution concentration during the quarantine period. However, the large decline observed in much of the world is projected to be exceeded again by the end of 2021 [46]. The “rebound effect” of the pandemic is estimated to produce an increase of 36.4 billion tons of carbon emissions from burning fossil fuels, an increase of 4.9% in 2021 compared with the previous year, reaffirming the continued reliance of the planet on carbon-based technologies.

Air quality is directly reflected in the third matrix evaluated, which refers to transportation. Clear decreases were observed in the use of the city’s BRT system, and the transition of transport modes must be evaluated. A notable proportion of people shifted from public to private transport, which is logical given the exposure to COVID-19 in mass transit modes. These findings are similar to those reported in a study carried out in Pakistan [47]. Travelers seem to prefer not to use public transport during the pandemic situation. Survey participants stated that they feared being infected with the virus while traveling on public transport, as there are chances of interacting with a person who is a carrier of the COVID-19 virus. Another study carried out in more than 15 countries in Asia, Europe, and North America found that most commuters have changed their trip habits. Shopping became the primary purpose for travel during COVID-19 [48].

This change might not be very beneficial if the percentage of people who start using vehicles or motorcycles increases to the extent that the vehicle fleet starts to have a greater impact on air pollution [49]. This situation may be prevented by the increase in the choice of active modes of transport, which generate enormous benefits given sustainable mobility.

We similarly observed an improvement in sewage water quality parameters. Reports from different regions around the world likewise show a reduction in water pollutants during the quarantine period [24]. This may be explained by the decrease in industrial water consumption. Although the report analyzed in this paper refers to the conditions in only one period of the two years of study, the decrease in parameters such as organic matter and turbidity represents a great relief for the city’s water bodies and their treatment logistics.

This study has many limitations. The first concerns the design of the study. Ecological studies look for associations between the occurrence of disease and exposure to known or suspected causes. However, because it is not possible to control different variables, biases can interfere with the analysis regarding the association between exposure and outcomes. Second, we did not include variables related to the effect of the pandemic on biodiversity due to the lack of data. Third, the data on changes in transport modes and perception of air quality are not representative of all cities. Finally, data were not available in the same format for all variables, i.e., the frequency of collection differed depending on the environmental matrix and source of the query, so data between matrices could not be compared to more accurately estimate the interrelationships among matrices.

## 5. Conclusions

This study showed the COVID-19 pandemic impact on the environment, offering an overview of changes in pollutants during 2019 and 2020 in Bogotá. The COVID-19 pandemic was a unique chance to analyze environmental impacts due to the fact that activities in large cities, such as Bogotá, were stopped over lockdown. The effects were both positive and negative. During the pandemic, Bogotá, like most cities around the world, adopted a strict lockdown to contain the spread of the virus. The months of strict restrictions generated significant changes associated with environmental and ecological conditions. The main air pollutants decreased markedly, attributed to travel restrictions by private cars or public transport.

This is relevant, considering that there is a long-term—potentially permanent—downward impact on the levels of environmental pressure, with stronger effects for pressure related to capital-intensive economic activities [50]. The promotion of telework as a result of lessons learned from the pandemic may positively impact some environmental matrices. Social experiments such as “a day without a car” have been undertaken in cities such as Paris, Berlin, Bogotá, and Toledo [51,52,53]. These experiences offer interesting results for improving mobility and air quality. More detailed studies are urged to understand and apply the lessons learned from the COVID-19 pandemic and its effects on the environment.

## Figures and Tables

**Figure 1 ijerph-19-06350-f001:**
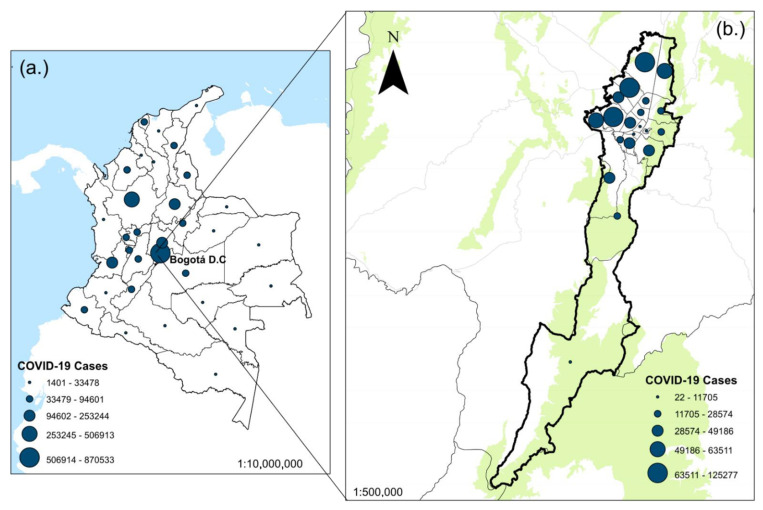
(**a**) SARS-CoV-2 cases in Colombia from 6 March 2020 until 18 May 2021; (**b**) study area, showing COVID-19 cases in Bogotá until 18 May 2021. The size of the circles is directly related to the number of infections since the first day of reporting. The geographical information system ArcGIS 10.5 for Desktop version 10.5.0.6491 was used to generate Figure 1. The Colombian administrative divisions and other geographic layers were downloaded from https://www.datos.gov.co/browse?sortBy=newest&utf8=%E2%9C%93 (accessed on 15 March 2022). COVID-19 data were obtained from Health Minister in Colombia, from 6 March 2020 to 18 May 2021.

**Figure 2 ijerph-19-06350-f002:**
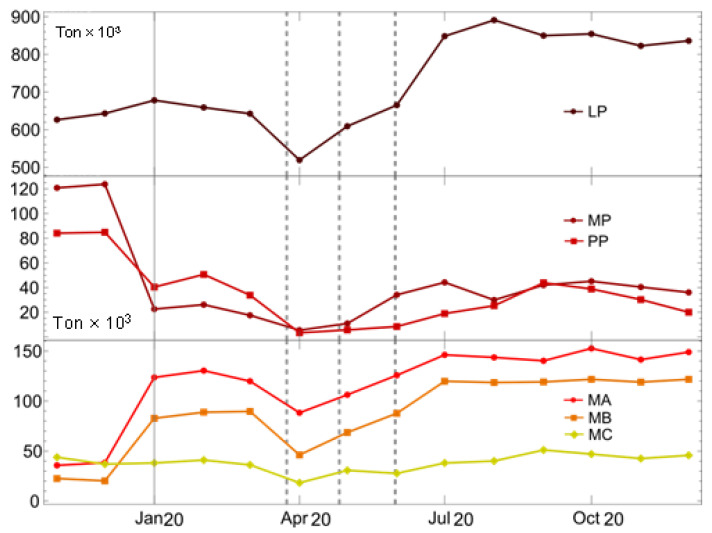
Bio-sanitary waste disposal series at the Bogotá sanitary landfill. The first panel shows biosanitary residues for the year 2020 for large producers (LPs), the second panel shows medium producers (MPs) and small producers (PPs), and the third panel shows micro-producers A, B, and C in red, orange and yellow, respectively.

**Figure 3 ijerph-19-06350-f003:**
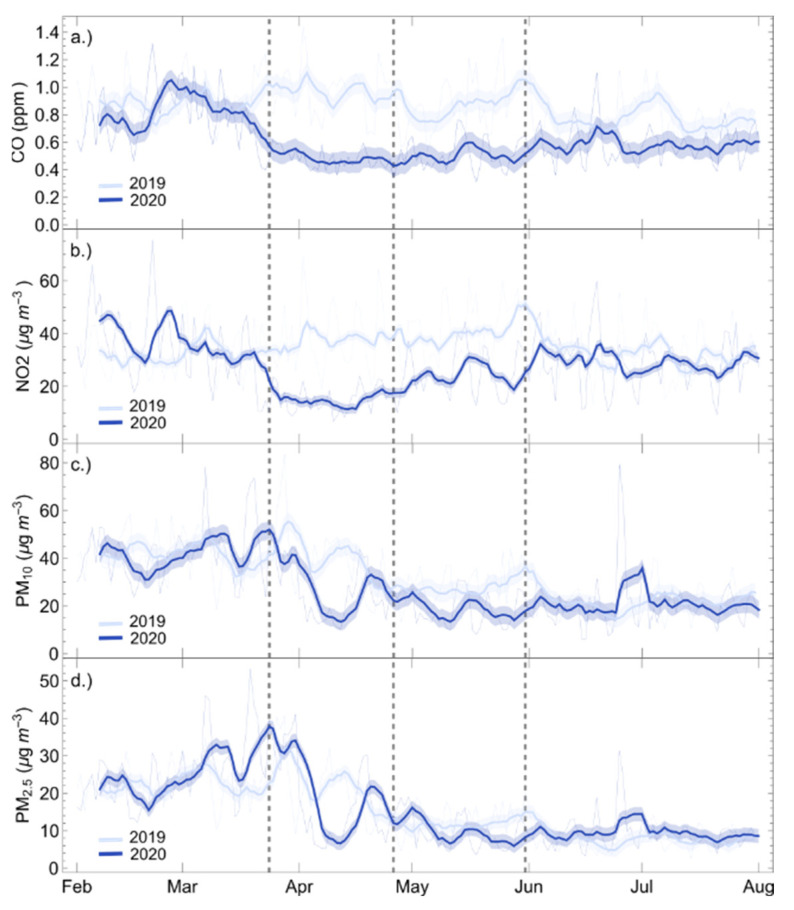
Air quality timeline: (**a**) daily average concentrations of carbon monoxide (CO); (**b**) nitrogen dioxide (NO_2_) concentrations; (**c**) the first panel corresponds to concentrations of particulate matter below 10 microns (PM_10_); (**d**) particulate matter below 2.5 microns (PM_2.5_). The light blue series shows moving concentrations in 2019 and the dark blue series shows concentrations in 2020.

**Figure 4 ijerph-19-06350-f004:**
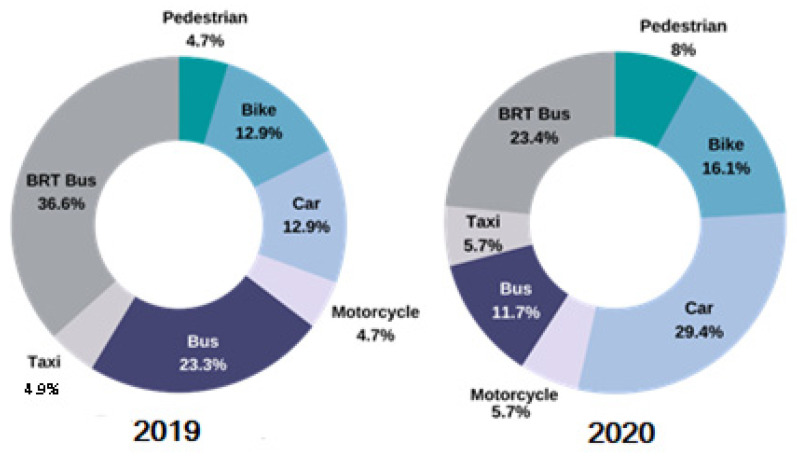
Transport modes in 2019 compared to 2020 or pandemic times according to the ITHACA perception survey.

## Data Availability

Not applicable.

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
