# Peer review of "Environmental Effects of the COVID-19 Pandemic: The Experience of Bogotá, 2020"

_ijerph, 2022, doi:10.3390/ijerph19106350_

Round 1

Reviewer 1 Report

Dear authors, 

I would like to thank you the opportunity of reviewing your paper. The issue is really interesting but, in my opinion, the current version of your manuscript should be revised in order to improve his quality:

  1. Theoretical backgrounds: It would be necessary to go in depth in the literature review. There are only 10 references in the introduction, and there is no specific section about it, so you should delve into the study of your topic and related to public health to justify more deeply the objectives of your research. I think it would not be a difficult task, due to the huge amount of literature existing. In addition, it could be interesting to detail what strategies have been followed in Colombia during the pandemic, restriction measures, quarantine period/s, etc. IJERPH it is an international journal, so it is really important to contextualize the specific situation in the country related to COVID-19 pandemic for any reader to understand its environmental impact.
  2. Discussion: According to my previous comment, this section should reflect the information described in the Introduction part, in both ways. For example, you talk about behavioural changes, lifestyles and habits related to the pandemic situation, but you have not mentioned these concepts in literature review before. On the other hand, as a recommendation, it would be more convenient to include limitations of the study in Conclusions section.
  3. Conclusions: This section is too short and could be completed. Moreover, it is necessary to point out the possible lines of work and reflecting what aspects have not could prove to drive future research.
  4. Formal issues: I strongly recommend the authors to adapt the size of some figures to fit them to the pages and make them easy and comprehensive to the readers.

I hope my comments would help you to improve your paper. Thanks and good luck.

Reviewer 2 Report

 The paper focuses a very interesting theme. The goal of the paper is clear and is well motivated.

The authors investigate the environmental factors of COVID-19 and present an overview of environmental matrices in Bogota. However, as acknowledged by the authors in the limitations part, the theoretical contribution of the paper is limited. The paper could offer more meaningful insights to the literature by strengthening the discussion and clearly emphasizing the differences between the present study and the previous works. The study is interesting, but both the paper structure and presentation need improvements to make it as a journal paper.

Reviewer 3 Report

I recommend the article for publication after incorporating the comments.

  • The article is clear and understandable. Appropriately and clearly describes the individual attributes of environmental impacts.
  • It is not clear how the companies were selected for the conclusions in section 3.1. Is the selection statistically significant? Please fill in.
  • The conclusions are too general. It would be appropriate to at least outline the dependencies between the individual attributes of the impacts.
  • It would be appropriate to unify the graphic outputs.
  • Proofreading is necessary.

Round 2

Reviewer 1 Report

Dear authors,

Thank you for making the suggested changes. The article is now ready for publication.

Congratulations and good luck!